

# Season of prescribed burns and management of an early successional species affect flower density and pollinator activity in a pine savanna ecosystem

Opeyemi A. Adedoja[1], Raelene M. Crandall[2] and Rachel E. Mallinger[1]

[1] Department of Entomology and Nematology, University of Florida, Gainesville, Florida, United States
[2] School of Forest, Fisheries, and Geomatics Sciences, University of Florida, Gainesville, Florida, United States

Corresponding author
Opeyemi A. Adedoja,
oadedoja@ufl.edu

## ABSTRACT

In the age of changing fire regimes, land managers often rely on prescribed burns to promote high diversity of herbaceous plants. Yet, little is known about how the timing of prescribed burns interacts with other ecological factors to maintain biodiversity while restoring fire-adapted ecosystems. We examined how timing of prescribed burns and removal of a dominant, early successional weedy plant yankeeweed (*Eupatorium compositifolium*) affect flower density and pollinator activity in an early-successional longleaf pine savanna restored from a timber plantation. During the first year of this study, plots received seasonal burn treatments, including unburned control, winter-dry, spring, and summer-wet season burns. During the second year of the study, data on flowers and pollinators were sampled across all plots. In the third year, these seasonal burn treatments were again applied to plots, and data were again collected on flowers and pollinators. In each burn treatment plot, we manipulated the presence of yankeeweed, including one control subplot (no removal) in which yankeeweed was not manipulated and one removal subplot in which yankeeweed was removed, and flowers and pollinators were measured. During the year between burns, flower density was highest in the summer-wet season burn treatment, significantly higher than in the unburned control, while pollinator activity was highest in the summer-wet and spring season burn treatments, significantly higher than the unburned control. During the year in which plots were burned again, flower density was highest in the spring season burn treatment, and pollinators most frequent in both spring and winter-dry season burn treatments, significantly higher than the unburned control. Removing yankeeweed enhanced pollinator activity but only in the year between fire applications.

We conclude that prescribed burning enhances floral resource availability and pollinator activity, but the magnitude of these effects depends on when fires are applied. Additionally, removal of yankeeweed can enhance pollinator activity during years between prescribed burns.

## INTRODUCTION

Prescribed burns play a significant role in the management of forested ecosystems in the southeastern United States and other ecosystems around the world (*Copeland, Sluis & Howe, 2002*; *Hartley et al., 2007*; *Smit et al., 2010*; *Adedoja et al., 2019*). Fires have been applied to prevent shrub encroachment and maintain herbaceous plant communities (*Platt, 1999*; *Lashley et al., 2014*), which in turn support diverse animal communities. The literature on the effects of prescribed burns on insect pollinators has expanded rapidly in recent decades and shows context-dependent and variable results. In general, prescribed burns initially decrease floral resources and insect pollinator populations but result in an increase 1–2 years after the burn (*Potts et al., 2003*; *Grundel et al., 2010*; *Mola & Williams, 2018*). In a long-term study in burned prairies, *Panzer (2002)* found both positive and negative responses to fire across 151 insect species, but of those species that responded negatively, 68% recovered within 1 year. This suggests that there are adverse, short-term effects of prescribed burns on some insect pollinators, but positive long-term effects on pollinator communities as a whole mediated by increased floral resources (*Van Nuland et al., 2013*).

The timing of burn can have varying effects on biological communities, with certain plant species having different survival and growth patterns between areas burned in the dry *vs.* wet seasons (*Main & Barry, 2002*; *Liu & Menges, 2005*; *Liu, Menges & Quintana-Ascencio, 2005*; *Baruzzi et al., 2022*). The timing of burn might have even more dramatic effects on insects by directly impacting them at different life stages and indirectly affecting them *via* changes in plant resource availability (*Brown et al., 2017*). For example, in prairie grasslands, *Johnson et al. (2008)* recorded 170% more insects in summer burns compared to winter burns, and similarly, bees were more abundant in summer-burned areas than in winter-burned areas (*Decker & Harmon-Threatt, 2019*). Collectively, these results suggest that prescribed burns lit during the historical, lightning-ignited fire season positively affect insect pollinator communities in prairie and other grassland ecosystems, but whether this is true for insect pollinator communities in forested ecosystems is largely unknown. Like many lightning fire-driven ecosystems, forested pine savannas are often fire excluded as a result of habitat fragmentation among other factors, requiring land managers to rely on prescribed burns that are often done outside of the historical lightning-ignited fire season in order to maintain the flora and fauna that characterize this ecosystem (*Ryan, Knapp & Varner, 2013*). Thus, understanding how the timing of prescribed burns mediates the role of fire as an effective management tool is critical for preserving pine savanna floral and pollinator communities.

In addition to prescribed burns, land managers often thin dominant vegetation in forested ecosystems to reduce the risks of high fire severity and its degrading effects on habitat composition and ecosystem functioning (*Dodson, Peterson & Harrod, 2008*; *Rossman et al., 2018*; *Ulyshen et al., 2022*). This is especially true in fire-driven ecosystems that have been modified by several years of fire exclusion and selective logging (*Harrod, McRae & Hartl, 1999*; *Merschel, Spies & Heyerdahl, 2014*; *Haugo et al., 2019*). Removal of dominant vegetation can also enhance herbaceous plant and flower diversity, which

benefits bees and other insect pollinators that thrive in open areas (*Taki et al., 2013*; *Rivers & Betts, 2021*). For example, mechanical shrub removal combined with prescribed burns in oak-dominated forests resulted in higher flower visitor activity and richness than either treatment alone, likely resulting from a higher percent herbaceous groundcover due to more sunlight reaching the understory (*Campbell, Hanula & Waldrop, 2007*). While some studies have assessed how thinning of dominant vegetation interacts with prescribed burns to maintain plant and pollinator communities (*Campbell, Hanula & Waldrop, 2007*; *Ulyshen et al., 2022*), we lack information on how the timing or season of prescribed burns mediates this relationship.

Yankeeweed, *Eupatorium compositifolium* Walter, is a native perennial species frequently described as an early successional or ruderal species and often perceived as a weed in the southeastern United States (*Macdonald et al., 1994*). It persists and dominates communities after soil disturbance for five or more years (*Grelen, 1962*; *Busing & Clebsch, 1983*), thus hindering restoration efforts in disturbed landscapes. Yankeeweed is managed with both chemical and mechanical removal (*Macdonald et al., 1994*) as it can grow to nearly 2 m in height and form significant shade over low-growing understory plants (*Macdonald, Brecke & Shilling, 1992*). Yankeeweed is characterized by anemophilous floral features, suggesting wind pollination as in closely related members of the *Eupatorium* genus (*Sullivan, 1975*; *Sullivan, Neigel & Miao, 1991*), but its pollination mechanism and value of its floral rewards to insects are unknown. Since yankeeweed can quickly recruit and become dominant after a soil disturbance, taking up most of the space, removing this species may influence plant community composition with implications for floral resource availability and pollinator activity.

We examined how both the timing of prescribed burns and removal of yankeeweed affect floral resources and insect pollinator activity in an early-successional pine savanna restoration. In the southeastern United States, land managers often apply prescribed burns during the winter-dry season (*Ryan, Knapp & Varner, 2013*) when many plants are dormant for ease and safety. However, wildfires historically occurred during the lightning season, which begins in spring and extends into the summer-wet season (*Fill, Davis & Crandall, 2019*). We initiated four seasonal burn treatments, including unburned control, winter-dry, spring, and summer-wet fires, and two yankeeweed treatments, including control and removal, and measured responses to our treatments by whole communities of plants and insect pollinators. We predicted that all seasonal burn treatments would result in greater flower density and diversity compared to unburned controls, but that effects would be most pronounced in areas burned in the spring and summer seasons because plants may have evolved with the historic lightning-season fire in this ecosystem. We also predicted that removing yankeeweed would result in a higher diversity of both floral resources and insect pollinators.

## METHODS

### Study design and site description

This study occurred at Austin Cary Forest (ACF) (29°43′N, 83°13′W), a 2,080-acre forest in Alachua County, Florida, USA. ACF was historically used as a timber and turpentine

operation before the University of Florida purchased the area in 1930; now, most of the land is used for research and teaching. ACF is managed with prescribed burns that vary in frequency and season.

The study plots were embedded in a 10-hectare early successional longleaf pine savanna. The broader study area is classified as a xeric upland pine savanna with predominantly Entisol soils. Upland pine savannas such as our study site are typically ignited every 2–5 years (*Huffman, 2006*). The entire area had previously been logged in 2013, burned in 2016, and planted in 2017 with longleaf pines, *Pinus palustris*. These restoration actions caused soil disturbance, which resulted in the dominance of yankeeweed, a perennial, ruderal species common in early successional fields. In late 2017, the area was divided into 12 plots and assigned to seasonal burn treatments, varying in the timing of prescribed burns, as described below. At the start of this study, the time since fire was identical across all plots, which had been burned in 2016 prior to initiating this experiment.

## Timing of prescribed burn treatments

We defined our burn treatments using precipitation patterns to easily compare our results with other study systems (*Fill, Davis & Crandall, 2019*; *Baruzzi et al., 2022*). Based on cumulative rainfall anomalies, which indicate trends of increasing and decreasing precipitation based on long-term averages, the timing of prescribed burn aligned with three distinct seasons: (1) winter, dry season burns hereafter referred to as winter-dry season burns; (2) late spring, early-wet season burns, hereafter referred to as spring burns; and (3) summer, wet season burns, hereafter referred to as summer-wet season burns. These three burn treatments and an unburned control were replicated three times across 12 plots. Treatments were randomly assigned to plots, with the exception that if random selection resulted in a treatment occurring in two adjacent plots, another treatment was randomly selected. Each burn treatment plot was approximately one hectare in size, but the exact plot size varied to allow for the placement of adequate fire breaks between plots. Burn treatment plots were directly adjacent to one another but with a fire break in between each plot. Seasonal burn treatments were first applied in 2018 prior to data collection. Winter-dry season burns were applied on March 22, 2018, spring burns on May 25, 2018, and summer-wet season burns on August 16, 2018. Data on flowers and pollinators were collected throughout 2019. Prescribed burns were conducted again in 2020, with winter-dry season burns applied on January 29, 2020, spring burns on June 18, 2020, and summer-wet season burns on August 9, 2020, and data were collected on flowers and pollinators in that same year (Fig. 1). For all seasonal burn treatments, backing fires were lit against the wind to establish a blackline before lighting successive strip fires across each plot. If subplots did not burn, the edges were relit with a head fire to ensure most of all plots were burned. Generally, summer-wet season burns consumed the most biomass with >95% of fuel burned, followed by winter-dry and spring season burns, which ranged from 75–95% fuel consumption across plots.

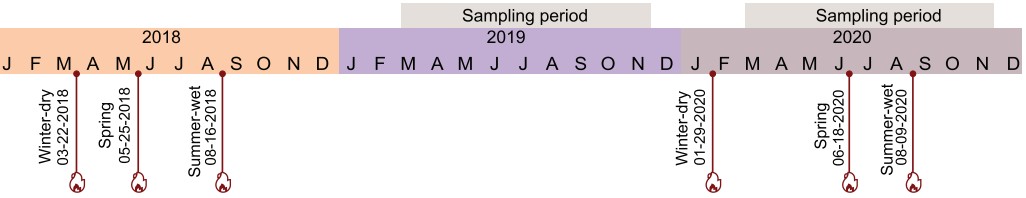

**Figure 1 Timeline of seasonal burn treatments and sampling period during which pollinators and flowering plants were recorded indicated by the grey bars.** The horizontal bars with different colors represent burn treatments in 2018, when burn treatments were first applied; 2019, the year between burn treatments, and 2020, when burn treatments were again applied. The marron vertical lines with fire symbols represent burn dates, and letters in each treatment year represent months of the year.

## Yankeeweed removal treatments

Within each fire treatment plot, two 10 m × 10 m yankeeweed treatment subplots were established, including one control subplot in which yankeeweed was not manipulated, hereafter referred to as the control plot, and one removal subplot in which yankeeweed was removed (~100% coverage reduction), hereafter referred to as the removal plot. Each subplot was located at least 10 m from plot borders. Within yankeeweed removal subplots and a 1 m buffer around the subplots, all yankeeweed plants were removed by clipping stems and their resprouts in late 2017–early 2018. In 2017, before applying yankeeweed removal treatments, the presence/absence and percent cover of plant species in each subplot were recorded, and no significant differences in plant community composition between subplots were found (*LaPierre, 2020*). Likewise, no differences in plant community composition among larger, burn treatment plots were found (*LaPierre, 2020*).

## Flower abundance and diversity

Flowers were sampled in each control and removal subplot per burn treatment plot in 2019 and 2020. Data collected in 2019 followed the first year of seasonal burn treatments (applied in 2018) and represented the year following fires in a typical biennial prescribed burn cycle, hereafter referred to as the year between prescribed burns or non-burn year (Fig. 1). Data collected in 2020 covered the second year following seasonal burn treatment initiation and represented a year in which fire was applied, hereafter referred to as the burn year (Fig. 1). Data were collected in all subplots once every 2 weeks, from early March to late November each year. Sampling every 2 weeks is frequent enough to capture differences in overall floral resource availability and flowering phenology across plant species (*Platt, Evans & Davis, 1988*). An observer recorded the flower density for plant species in bloom in each 10 m × 10 m subplot during each sampling event. For all non-composite flowering plant species, individual flowers were counted. Each flower head was counted as one unit or flower for composite flowering plant species (*e.g.*, Asteraceae). For plant individuals with numerous (~1,000+) flowers, flower abundance was estimated by counting plant stems in bloom and multiplying by the average number of flowers per 10 stems per plant. Most plants were recorded to species (60/64), with a minority recorded to the genus level (4/64). Species richness of plants in bloom on each sampling date was determined as the number of species blooming per subplot.

## Pollinator activity and diversity

Sampling for insect pollinators occurred from 8:00–13:00 when bees and other pollinators forage (*Danforth et al., 2019*). Observations only took place under partial to full sun, with no precipitation and temperatures 13 °C or greater. Due to the proximity of the treatment plots to one another, all within a 10-hectare area, we do not assume that treatment plots host independent pollinator communities given the large foraging range of many insect pollinators (*Goulson & Stout, 2001*; *Pasquet et al., 2008*). Thus, sampling was designed to measure differences in pollinator activity. To measure pollinator activity, pollinators were counted during 5-min timed observations per subplot on each sampling day in which flower density was also recorded. Two observers starting on opposite sides of the study area walked around each subplot in a slow, meandering manner for 5-mins per subplot while being careful not to disturb plants. In addition to standardized 5-min observation periods, observers also opportunistically recorded any insects seen on flowers during flower counts as described above. All insects were recorded to the following categories representing functional groups and/or broad taxonomic categories: honeybees (*Apis mellifera*), bumble bees (*Bombus* spp.), other bees (non-*Apis*, non-*Bombus* Apoidea, primarily solitary bees), wasps, flies (Diptera), butterflies and moths (Lepidoptera), and crawling insects (including ants and beetles). In addition to recording all pollinators to these broad categories, observers recorded the lowest identifiable taxonomic level for each insect. The order in which burn treatment plots were sampled changed across sampling days, and two observers working simultaneously ensured that all subplots within all burn treatment plots were sampled within a ~3-h period.

Flower visitors were not captured during the 5-min observation periods for the following reasons, which generally would have affected our ability to accurately assess the main response variable of interest, pollinator activity: (1) netting insects on low-lying plants is likely to disrupt other visitors in the area, (2) collecting insect visitors on the first observation of their foraging would limit latter observations of these same insects, (3) across such broad taxa of insects, some are much more readily captured than others, and (4) flower visitors were generally infrequent and thus lethal capturing was kept to a minimum. Instead, after standardized 5-min observation periods, in subplots where insect activity was observed, observers opportunistically collected a subset of insects from novel plant-pollinator interactions to identify them to the species level. As such, collecting did not have a standardized intensity but was instead done to gather descriptive information on plant-pollinator interactions in this ecosystem. All collected insects were pinned and identified using Discover Life, a reference collection of wild bees identified by entomologist Glenn Hall and taxonomist John Ascher, and an online key to bees of Florida (*Pascarella & Hall, 2016*).

## Statistical analyses

We used generalized linear mixed effects models to examine the effects of timing of prescribed burn and yankeeweed removal treatment on flower density, flower richness, and pollinator activity (*i.e.*, visitation rates). Models included the following fixed effects: seasonal burn treatment, yankeeweed removal treatment, day of year, interactions between

seasonal burn treatment and day of year and between yankeeweed removal treatment and day of year, year (2019, 2020), interactions between seasonal burn treatment and year and between yankeeweed removal treatment and year, and time-since-burn (continuous, in days). In another model, we removed the density of yankeeweed from the overall flower density and used all fixed effects described above for other models, but also included a three-way interaction between seasonal burn treatment, yankeeweed removal treatment, and year to assess how the effect of yankeeweed treatment on flower density (excluding yankeeweed) varies across burn treatments and years. All models also included a random intercept of subplot (1–24) nested within plot (1–12) to account for correlated errors among subplots within plots repeatedly sampled within a year. We compared models with a Poisson distribution and a negative binomial distribution for each response variable and chose the best fit model (lowest AICc), which used a Poisson distribution for flower density and richness, and a negative binomial distribution for pollinator activity. Models were created with the "glmer" function using the *lme4* package (*Bates et al., 2014*) in R. Statistical significance for each fixed effect was determined with Type II sum of squares ANOVA when interactions were not significant and Type III sum of squares when they were significant (see *Lewsey, Gardiner & Gettinby, 2001*; *Langsrud, 2003*).

We additionally examined year-long, cumulative richness of flowers using linear mixed models with the following fixed effects: seasonal burn treatment, yankeeweed removal treatment, year (2019, 2020), the interaction between seasonal burn treatment and year, and an interaction between yankeeweed removal treatment and year. Residuals of the year-long flower richness per subplot were normally distributed. Estimated marginal means and 95% confidence intervals for fire and yankeeweed treatments, and their respective interactions, were calculated using the *effects* package in R, and pairwise comparison of estimated marginal means were performed using the *emmeans* package (*Lenth et al., 2021*) in R. All residual diagnostics were computed using the *DHARMa* package (*Hartig, 2018*).

To examine the effects of treatments on flower community composition each year, we constructed a Bray-Curtis dissimilarity matrix using the year-long proportion of each flower species in each subplot. For this analysis, as the goal was to examine how yankeeweed removal influences the remaining community composition of floral resources, we removed yankeeweed from the community in each subplot. We calculated the proportions of the remaining total floral resources and ran a PERMANOVA on the Bray-Curtis dissimilarity matrix with the following explanatory variables: seasonal burn treatment, yankeeweed removal treatment, year (2019, 2020), the interaction between seasonal burn treatment and year, and an interaction between yankeeweed removal treatment and year. For significant effects, we followed the PERMANOVA with a SIMPER analysis to identify flower species that significantly contribute to differences between treatments. Analyses were done using the R package *vegan* (*Oksanen et al., 2017*).

Finally, we constructed plant-pollinator networks for each year using data combined across all plots and subplots using the R package *igraph* (*Csardi & Nepusz, 2006*). The network was created with weighted matrices containing interaction frequencies between each flowering plant species and pollinator taxa. For simplicity, we show all

pollinator taxa but only plant taxa that received pollinator visits. Data informing these networks came from both timed observations and opportunistic pollinator collections across all plots and subplots.

## RESULTS

### Plant-pollinator interactions in this ecosystem

We recorded 64 and 69 flowering plant species in 2019 and 2020, respectively. *Eupatorium compositifolium* (yankeeweed), *Dalea pinnata*, *Callicarpa americana*, and *Croton michauxii* were the most abundant flowers in both years, although *D. pinnata* had fewer flowers in 2020. Insects visited a total of 39 and 46 flowering plant species across all subplots and sampling dates in 2019 and 2020, respectively (Appendix 1). In both years, insect visits were most frequent to the following plant species: *C. americana*, *Chamaecrista fasciculata*, *Croptilon divaricatum*, *C. micauxii*, *D. pinnata*, *Rubus cunefolius*, and *Vaccinium stamineum*. Insects visited more flowering plants in 2020, including the following additional species: *E. compositifolium*, *Indigofera* sp., *Liatris* sp., *Solidago* spp., *Serenoa repens*, and *Stillingia sylvatica* which received few to no pollinator visits in 2019. *Dalea pinnata* attracted honeybees, butterflies, and flies primarily, and *C. americana* attracted mostly bees and wasps. Additionally, *E. compositifolium* attracted primarily flies, and *S. repens* attracted mainly crawling visitors and flies. Other plants such as *Liatris* sp., *C. fasciculata*, *C. michauxii*, *C. divaricatum*, and *Indigofera* sp. attracted a variety of insects, including a diversity of bees (Appendix 1).

### Effects of prescribed burn treatments on flowers and pollinators

Prescribed burn treatments had a significant effect on flower density, but this effect varied across sampling dates as indicated by a significant interaction (Table 1). Flower density generally increased throughout the season and peaked in October during wildflower bloom; differences in flower density across burn treatments also increased throughout the season and were most pronounced during peak bloom (Table 1). Furthermore, prescribed burns had varying effects on flower density across sampling years; in 2019 when plots were not burned (*i.e.*, in between fire treatments), the summer-wet season fire treatment plots had the highest flower density, significantly higher than unburned control plots (Table 1; Fig. 2A), while in 2020 when plots were burned for a second time, the highest flower density was observed in the spring season fire treatment plots, significantly higher than unburned control plots (Table 1; Fig. 2A).

Flower richness did not vary significantly among burn treatments when analyzed per sampling date or as total richness per year (Table 1). Flower community composition also did not differ significantly among burn treatments (Table 1), and this was consistent across years (Table 1). However, flower communities did vary between years (Table 1). In 2019, *D. pinnata*, *E. canadensis*, and *C. divaricutum* were relatively more abundant, while in 2020, *C. michauxii*, *S. repens*, *V. stamineum*, *C. americana*, and *Solidago* spp. were relatively more abundant (Table 2).

The effect of prescribed burns on pollinator activity varied across years (Table 1; Fig. 2B) but was consistent across sampling dates within a year (Table 1). In 2019,

**Table 1 Model summary table for flower density, flower density excluding yankeeweed, flower richness, pollinator activity, cumulative flower richness per year, flower community composition, predicted as a function of fire season treatment (FT), yankeeweed treatment (YT), time since fire (TSF), day of year (DOY), and year.** Test statistics value represents output of generalized linear mixed-effect model (GLMM) or linear mixed-effect models (LMM) or permutation analysis of variance (PERMANOVA). Estimated means and 95% CI (2.5% and 97.5% percentiles) around the means were calculated for treatment combinations of nominal variables using the effects package in R. Parameter estimates and 95% CI were calculated for all other treatments and their interactions with continuous variables (DOY and TSF).

| Response variable | Explanatory variable | df | Mean/parameter estimates | Lower 95% CI | Upper 95% CI | Test statistic | P value |
|---|---|---|---|---|---|---|---|
| GLMM | | | | | | $(\chi^2)$ | |
| Flower density | FT * DOY (Estimates) | 3 | | | | 5,555 | <0.0001 |
| | *Unburned*DOY* | – | −0.11 | −0.85 | 0.63 | – | – |
| | *Winter-dry*DOY* | – | −0.53 | −1.23 | 0.16 | – | – |
| | *Spring*DOY* | – | −0.20 | −0.89 | 0.49 | – | – |
| | *Summer-wet*DOY* | – | 0.11 | −0.61 | 0.83 | – | – |
| | FT * Year (Mean) | 3 | | | | 341,270 | <0.0001 |
| | **2019** | | | | | | |
| | *Unburned* | – | 533.19 | 331.16 | 858.47 | – | – |
| | *Winter-dry* | – | 482.76 | 285.75 | 815.59 | – | – |
| | *Spring* | – | 581.03 | 334.60 | 979.67 | – | – |
| | *Summer-wet* | – | 1,110.16 | 647.08 | 1,904.65 | – | – |
| | **2020** | | | | | | |
| | *Unburned* | – | 439.59 | 272.99 | 707.87 | – | – |
| | *Winter-dry* | – | 689.85 | 408.32 | 1,165.50 | – | – |
| | *Spring* | – | 1,140.94 | 676.65 | 1,923.84 | – | – |
| | *Summer-wet* | – | 368.68 | 214.88 | 632.59 | – | – |
| | TSF (Estimates) | 1 | 0.000073 | 0.000055 | 0.000091 | 65.26 | <0.001 |
| | YT*DOY (Estimates) | 1 | −0.011 | −0.0094 | −0.082 | 115,120 | <0.001 |
| | YT*Year (Mean) | 1 | | | | 899.25 | <0.001 |
| | **2019** | | | | | | |
| | *Control* | – | 744.71 | 516.80 | 1,073.14 | – | – |
| | *Removal* | – | 547.16 | 374.52 | 799.38 | – | – |
| | **2020** | | | | | – | – |
| | *Control* | – | 728.61 | 505.61 | 1,049.95 | – | – |
| | *Removal* | – | 490.20 | 335.53 | 719.16 | – | – |
| Flower density excluding yankeeweed | FT*YT*Year (Mean) | 3 | | | | 4,759.50 | <0.001 |
| | **Unburned 2019** | | | | | | |
| | *Control* | – | 176.26 | 83.71 | 371.13 | – | – |
| | *Removal* | – | 285.42 | 134.34 | 605.98 | – | – |
| | **Winter-dry 2019** | | | | | | |
| | *Control* | – | 173.01 | 81.29 | 368.19 | – | – |
| | *Removal* | – | 619.74 | 290.41 | 1,322.52 | – | – |
| | **Spring 2019** | | | | | | |
| | *Control* | – | 477.62 | 222.95 | 1,023.19 | – | – |
| | *Removal* | – | 749.99 | 350.37 | 1,605.36 | – | – |

(Continued)

| Table 1 (continued) | | | | | | | |
|---|---|---|---|---|---|---|---|
| **Response variable** | **Explanatory variable** | **df** | **Mean/parameter estimates** | **Lower 95% CI** | **Upper 95% CI** | **Test statistic** | **P value** |
| | **Summer-wet 2019** | | | | | | |
| | *Control* | – | 392.71 | 184.27 | 836.88 | – | – |
| | *Removal* | – | 505.15 | 236.82 | 1,077.55 | – | – |
| | **Unburned 2020** | | | | | | |
| | *Control* | – | 168.64 | 80.08 | 355.14 | – | – |
| | *Removal* | – | 200.82 | 94.57 | 426.45 | – | – |
| | **Winter-dry 2020** | | | | | | |
| | *Control* | – | 339.37 | 159.47 | 722.24 | – | – |
| | *Removal* | – | 263.58 | 123.49 | 562.58 | – | – |
| | **Spring 2020** | | | | | | |
| | *Control* | – | 262.31 | 226.34 | 562.02 | – | – |
| | *Removal* | – | 247.94 | 105.71 | 530.77 | – | – |
| | **Summer-wet 2020** | | | | | | |
| | *Control* | – | 482.34 | 122.43 | 1,027.88 | – | – |
| | *Removal* | – | 225.51 | 115.82 | 481.10 | – | – |
| Flower richness | FT * DOY (Estimates) | 3 | | | | 0.93 | 0.82 |
| | *Unburned*DOY* | – | −0.175 | −0.620 | 0.271 | – | – |
| | *Winter-dry*DOY* | – | 0.226 | −0.238 | 0.690 | – | – |
| | *Spring*DOY* | – | 0.090 | −0.355 | 0.534 | – | – |
| | *Summer-wet*DOY* | – | 0.175 | −0.271 | 0.620 | – | – |
| | YT*DOY (Estimates) | 1 | −0.00069 | −0.0018 | 0.00038 | 1.60 | 0.21 |
| | TSF (Estimates) | 1 | 0.00036 | 0.000067 | 0.00065 | 5.79 | 0.02 |
| | YT (Estimates) | 1 | 0.19 | −0.19 | 0.49 | 0.0043 | 0.95 |
| Pollinator activity | FT * DOY (Estimates) | 3 | | | | 7.23 | 0.07 |
| | *Unburned*DOY* | – | −0.71 | −1.96 | 0.54 | – | – |
| | *Winter-dry*DOY* | – | 1.52 | 0.14 | 2.89 | – | – |
| | *Spring*DOY* | – | 0.69 | −0.55 | 1.94 | – | – |
| | *Summer-wet*DOY* | – | 0.71 | −0.54 | 1.96 | – | – |
| | FT * Year (Mean) | 3 | | | | 9.45 | 0.02 |
| | **2019** | | | | | | |
| | *Unburned* | – | 0.75 | 0.37 | 1.50 | – | – |
| | *Winter-dry* | – | 1.71 | 1.04 | 2.79 | – | – |
| | *Spring* | – | 2.61 | 1.58 | 4.32 | – | – |
| | *Summer-wet* | – | 3.20 | 1.90 | 5.40 | – | – |
| | **2020** | | | | | | |
| | *Unburned* | – | 1.47 | 0.56 | 3.82 | – | – |
| | *Winter-dry* | – | 9.56 | 5.22 | 17.50 | – | – |
| | *Spring* | – | 6.99 | 4.22 | 11.58 | – | – |
| | *Summer-wet* | – | 5.18 | 3.17 | 8.48 | – | – |
| | TSF (Estimates) | 1 | 0.0011 | 0.00021 | 0.0021 | 5.77 | 0.02 |

| Response variable | Explanatory variable | df | Mean/parameter estimates | Lower 95% CI | Upper 95% CI | Test statistic | P value |
|---|---|---|---|---|---|---|---|
| | **Table 1 (continued)** | | | | | | |
| | YT*Year (Mean) | 1 | | | | 5.21 | 0.02 |
| | **2019** | | | | | | |
| | *Control* | – | 1.33 | 0.95 | 1.87 | – | – |
| | *Removal* | – | 2.46 | 1.76 | 3.44 | – | – |
| | **2020** | | | | | | |
| | *Control* | – | 4.73 | 3.42 | 6.53 | – | – |
| | *Removal* | – | 4.77 | 3.46 | 6.58 | – | – |
| LMM | | | | | | (χ²) | |
| Cumulative flower richness per year | FT (Estimates) | 3 | | | | 3.51 | 0.32 |
| | *Unburned* | | 1.33 | −5.33 | 7.99 | – | – |
| | *Winter-dry* | | 2.17 | −4.49 | 8.83 | – | – |
| | *Spring* | | −1.50 | −8.16 | 5.16 | – | – |
| | *Summer-wet* | | −1.33 | −7.99 | 5.33 | – | – |
| | YT | 1 | −0.50 | −5.21 | 4.21 | 0.41 | 0.52 |
| PERMANOVA | | | | | | F | |
| Flower composition | FT | 3 | – | – | – | 1.24 | 0.23 |
| | Year | 1 | – | – | – | 4.72 | 0.003 |
| | FT*Year | 3 | – | – | – | 0.38 | 0.99 |

pollinator activity was highest in the summer-wet and spring season fire treatment plots, both significantly higher than in unburned control plots (Table 1). In 2020 when fire treatments were again applied, pollinator activity was highest in the winter-dry and spring season burn treatments, both significantly higher than the unburned control plots (Table 1; Fig. 2B).

After accounting for the effects of prescribed burn treatments, time-since-fire had a positive effect on both flower density and flower richness as well as on pollinator activity (Table 1).

## Effects of yankeeweed removal on flowers and pollinators

The effect of yankeeweed removal on flower density varied across sampling dates (Table 1) and years (Table 1; Fig. 3A). Overall, flower density was higher in control (no removal) plots in both years (Fig. 3A), and the blooming of yankeeweed largely drove this result. Yankeeweed removal did not affect flower richness per date (Table 1), total flower richness per year (Table 1), or flower community composition (Table 1). When we analyzed the effect of removal treatment on the flower density of all other plants (excluding yankeeweed), the effect of yankeeweed removal on flower density varied across burn treatments and years as indicated by the three-way interaction (Table 1; Fig. 4). In 2019, flower density was higher in yankeeweed removal plots compared to control plots, but only

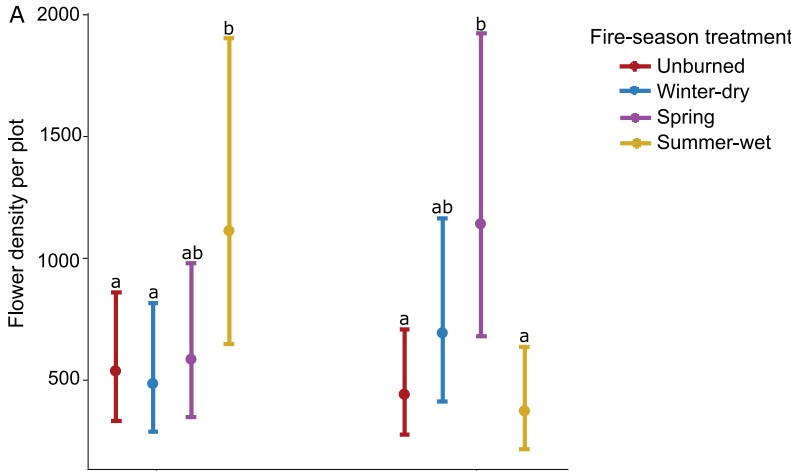

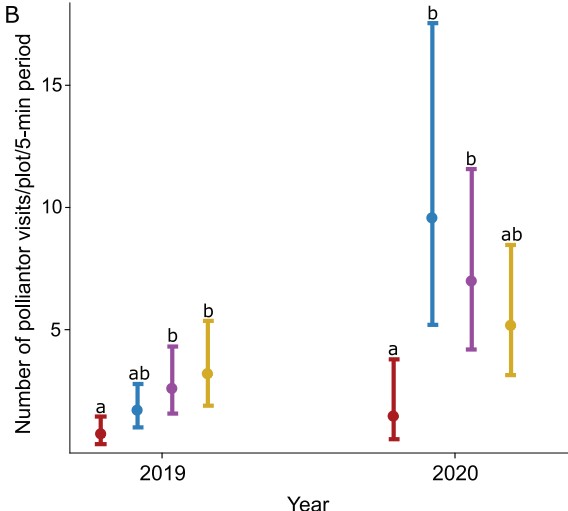

**Figure 2 Effect of seasonal burn treatments on (A) flower density and (B) pollinator activity in 2019, the year between burn treatments, and 2020, when burn treatments were again applied.** Plotted values represent estimates from generalized linear mixed effect models with (A) Poisson error for flower density and (B) negative binomial error for pollinator visitation rates. Bars represent 95% confidence intervals around the estimates. Bars with similar letters in each year are not significantly different at alpha = 0.05.

in the winter-dry season burn treatment (Fig. 4; Table 1). In 2020, flower density was less variable between yankeeweed removal and control plots (Fig. 4).

The effect of yankeeweed removal on pollinator activity varied across years (Table 1), with higher pollinator activity recorded in the yankeeweed removal plots in 2019 but comparable frequencies between treatments in 2020 (Fig. 3B).

## DISCUSSION

While fire remains a significant driver of plant and pollinator distributions, understanding how other environmental factors mediate the role of fire as a management tool may aid conservation efforts to ameliorate current trends of pollinator decline.

**Table 2 Plant taxa contributing to differences in plant community composition between 2019 (a non-burn year) and 2020 (a burn year) in a pine savanna restoration site.** "Ratio" is the ratio of the plant's average contribution to overall dissimilarity relative to the standard deviation of its contribution. $\bar{X}$ is the average proportion of the plant within the community each year. $\Sigma$ Cum is the ordered cumulative contribution of the plant. Plants are listed in order of contribution until a cumulative contribution of ≥75% was reached. Results were obtained from a similarity percentages (SIMPER) analysis.

| Plant taxa | Ratio | $\bar{x}$ 2019 | $\bar{x}$ 2020 | $\Sigma$ Cum |
|---|---|---|---|---|
| *Croton michauxii* | 1.34 | 0.289 | 0.392 | 0.22 |
| *Dalea pinnata* | 1.09 | 0.284 | 0.032 | 0.428 |
| *Callicarpa americana* | 0.94 | 0.117 | 0.123 | 0.548 |
| *Erigeron canadensis* | 1.20 | 0.087 | 0.052 | 0.606 |
| *Croptilon divaricatum* | 1.10 | 0.072 | 0.061 | 0.660 |
| *Serenoa repens* | 0.55 | 0.006 | 0.056 | 0.705 |
| *Solidago* spp. | 0.51 | 0.025 | 0.038 | 0.749 |
| *Vaccinium stamineum* | 0.52 | 0.018 | 0.040 | 0.788 |

Our replicated, randomized experimental approach, followed by frequent data collection, facilitated a unique assessment of how plant and pollinator communities changed over time in response to prescribed burns. We predicted that burning would increase flower density and diversity, and that removing a dominant early-successional plant species—yankeeweed—would further enhance flower and pollinator diversity. In general, while burning positively affected flower density and pollinator activity, the timing of fire mediated this effect. In the year following a burn, flower density and pollinator activity were highest in plots that had previously been burned in the summer-wet and spring seasons, significantly higher than unburned plots and in line with our predictions that fires applied during the lightning season have the strongest positive effects on plant and pollinator communities. In the year of a burn, however, flower densities were highest in the spring season fire treatment, while pollinator activity was highest in the winter-dry and spring season fire treatments. Yankeeweed removal had positive effects on pollinator activity, but only in the year following a burn, and did not have consistent effects on the density of other flowers. These results illustrate differential effects of fire timing and yankeeweed removal across years of a burn cycle but generally support the use of fire and weedy plant removal to maximize pollinator and floral resource density and diversity.

In the southeastern United States, wildfires were once frequent during the lightning season (late spring-summer); however, land managers often apply prescribed burns in the winter season when many plants are dormant (*Ryan, Knapp & Varner, 2013*). Because we found that the winter-dry season burn treatment had low flower densities in both the burn and non-burn years, comparable to unburned control plots, our results suggest that burning in the dry season does not optimize floral resources for pollinators as compared to burning in the spring or summer when fires historically occurred. The peak bloom period of many herbaceous plant species in the southeastern United States is in late summer and fall (September to November). Thus, prescribed burns in the summer may disrupt the flowering of many species as fire is applied immediately before peak bloom season. Still, it

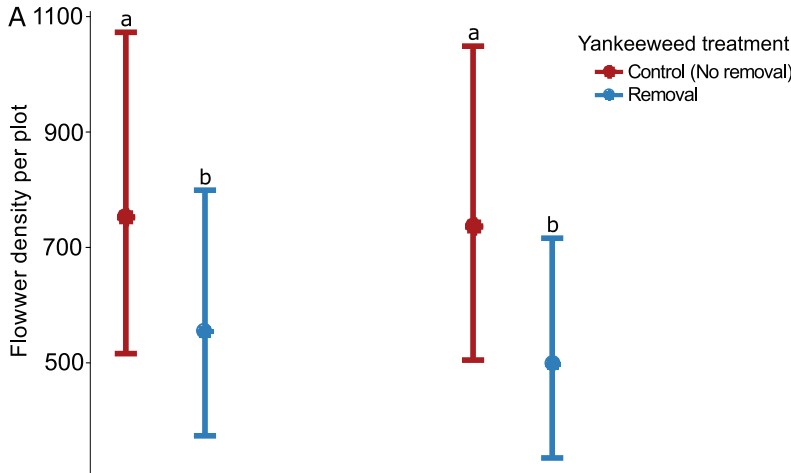

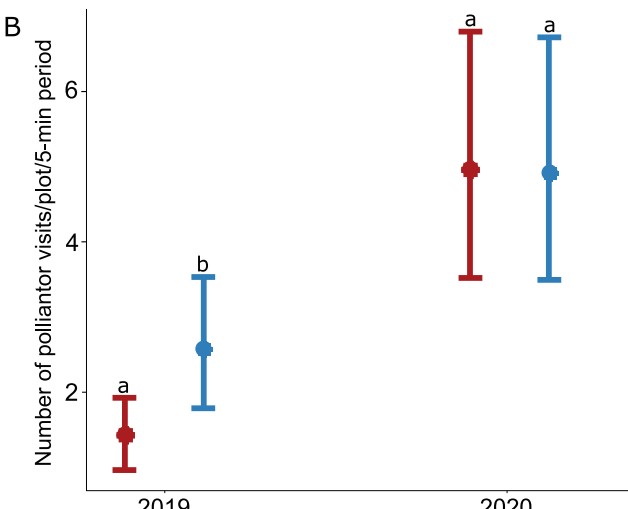

**Figure 3 Effect of yankeeweed removal treatments on (A) flower density, and (B) pollinator visitation rates in 2019, the year between burn treatments, and 2020, when burn treatments were again applied.** Plotted values represent estimates from generalized linear mixed effect models with (A) Poisson error for flower density and (B) negative binomial error for pollinator visitation rates. Bars represent 95% confidence intervals around the estimates. Bars with similar letters in each year are not significantly different at alpha = 0.05.     

may enhance the flowering of these same species in the year following a burn. This may explain the low and high flower densities recorded in the summer-wet season fire treatments in the burn and non-burn years, respectively. On the other hand, burning in the spring well in advance of the late summer-fall bloom may enhance the blooming of these species during a burn year, explaining the high flower density recorded in the spring season fire treatment during the burn year. Thus, understanding the interaction of prescribed burn timing with plant phenology is key when evaluating which burn season is optimal for

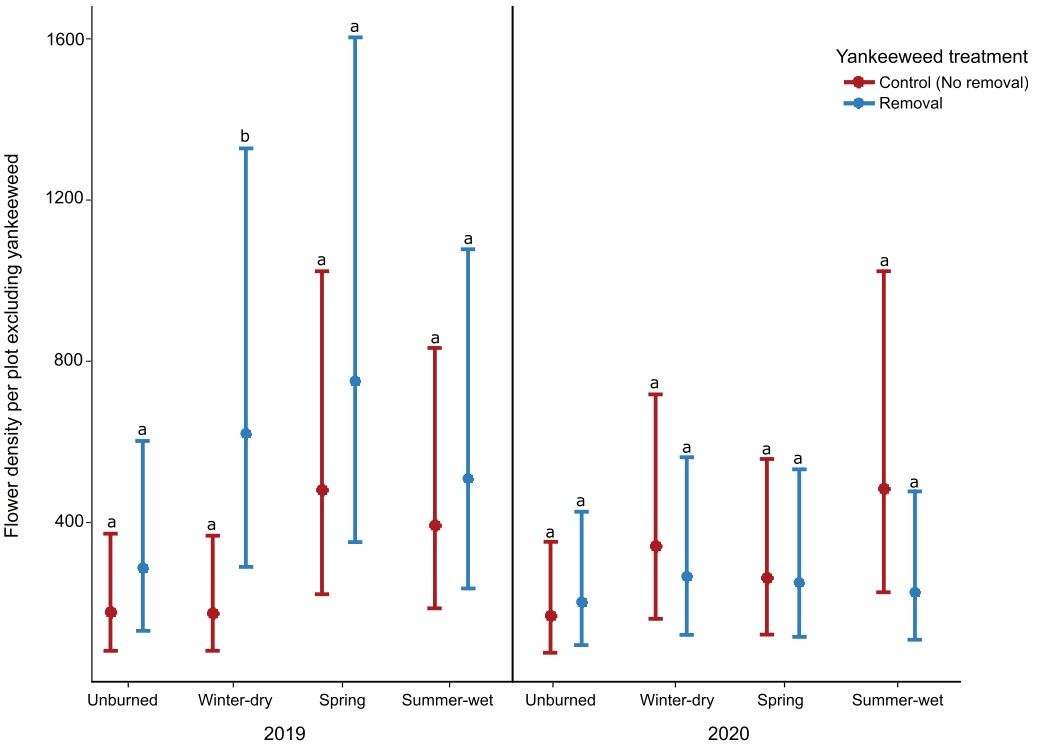

**Figure 4 Effect of yankeeweed removal treatments on flower density excluding yankeeweed across seasonal burn treatments in 2019, a year between burn treatments, and in 2020, a year in which burn treatments were again applied.** Plotted values represent estimates from generalized linear mixed effect-model three-way interactions with Poisson error. Bars represent 95% confidence intervals around the estimates. Bars with similar letters in each year are not significantly different at alpha = 0.05.

promoting high floral density in fire-adapted ecosystems (*Platt, Evans & Davis, 1988*; *Knapp, Estes & Skinner, 2009*; *Pavlovic, Leicht-Young & Grundel, 2011*).

While our results show that fire affects flower density, this was not mediated by differences in plant species composition, as plant communities did not vary significantly across burn treatments. Although plant communities did not vary across burn treatments, plant communities did differ between the burn and non-burn years; most of the plant species that flowered in greater densities in burned plots or only in the burn year have fire-related traits. For example, *Serenoa repens* resprouts following fires that remove aboveground tissues (*Carrington & Mullahey, 2006*), while *Liatris* spp. depend on chemicals from smoke to cue germination (*Lindon & Menges, 2008*), and both species have been recorded as having relatively greater flowering densities during burn years. In fire-adapted communities, enhanced post-fire floral diversity has been attributed to fire-triggered flowering and a general increase in resource availability for some plant species (*Brewer & Platt, 1994*; *Fill, Davis & Crandall, 2019*; *Wagenius, Beck & Kiefer, 2020*). Although the time scale on which plants respond to fire may vary among species, thus explaining varying floral densities between burn and non-burn years, fire in general creates more suitable conditions for flowering understory vegetation to thrive through increased soil nutrients (*Dudley & Lajtha, 1993*), reduced competition for biotic and abiotic

resources (*Knapp, Estes & Skinner, 2009*), and increased access to sunlight (*Bourg, Gill & McShea, 2015*). These factors also help maintain pollinator communities (*Burkle et al., 2019*), as pollinators depend on floral resource availability, which generally increased with fire in this study.

Pollinator activity generally followed the same patterns as flower density, at least in the year following a burn, and was highest in the summer-wet season and spring season burn treatment plots. Since pollinator activity is largely driven by flower density (*Westphal, Steffan-Dewenter & Tscharntke, 2003*; *Vrdoljak, Samways & Simaika, 2016*), it is not surprising that we recorded similar patterns of pollinator activity and flower density across treatments. However, in the year of the burn, pollinator activity did not follow trends in flower density as closely; higher pollinator activity was recorded in winter-dry season burn plots compared to unburned control plots, but flower density was comparable between winter-dry season and unburned plots. It is thus unclear what is driving relatively higher pollinator activity in winter-dry season burn treatment plots, but potential mechanisms include increased production of nectar per flower or earlier emergence of insects in that treatment following a burn (*Ratnieks & Balfour, 2021*). Overall, however, across both years, our results support the use of fire to enhance floral resources and pollinator activity, as certain burn treatments showed greater floral resources and pollinator activity than unburned controls. In addition to the indirect effect of fire on pollinators through increasing floral density, fire directly provides more bare ground and woody debris for ground and cavity nesters, respectively, though these benefits may be primarily seen in years following fires (*Burkle et al., 2019*).

While fire generally had a positive effect over time compared to unburned controls, we also found a positive effect of time since fire on both flower density and pollinator activity. As this effect was significant after accounting for the overall burn treatment effect, as well as overall differences between the two sampling years, it generally shows the immediate, negative impact of fire on both flowers and pollinators within a given treatment and year. This immediate, negative impact was generally short-lived, as even within a burn year, flower density and pollinator activity reached high levels in the weeks following the application of a burn.

Yankeeweed was the most dominant plant in our study, and thus its removal reduced overall flower density. However, removing yankeeweed can improve floral resource diversity and pollinator recruitment; yankeeweed removal was associated with higher pollinator activity, at least in non-burn years. Its removal also resulted in a greater density of other floral resources, though this was only significant in the winter-dry season burn treatment in the year following a burn. Although yankeeweed is native to this region, it spreads rapidly and dominates human-disturbed landscapes. Thus, controlling it can potentially aid habitat recovery during restoration. Specifically, mechanical manipulation of yankeeweed along with prescribed burns may aid in the long-term recovery of co-existing pollinator plants. Since yankeeweed is taller than most other pine savanna understory species, reducing it may increase sunlight reaching the understory, thereby increasing flowering plant density (*Adedoja, Kehinde & Samways, 2021*). However, manipulating yankeeweed may require a careful approach to reduce soil disturbance, since

this plant persists and spreads rapidly after soil disturbance (*Grelen, 1962*; *Busing & Clebsch, 1983*). Also, yankeeweed produces a copious number of flowers and is attractive to insects, in particular flies, showing that it may offer some floral rewards for flower visitors (*Atwater, 2013*). Controlling yankeeweed could be important for restoring pollinator community and diversity of floral resources in disturbed areas, but the complete removal of yankeeweed may threaten the long-term preservation of this native species.

# CONCLUSION

Many ecologists and land managers rely on prescribed burning to conserve biodiversity; however, other factors mediating the impact of prescribed burns are often neglected. Our multi-year study shows that prescribed burns generally enhance pollinator activity and flower density, but the role of fire as an effective management tool for maintaining floral and pollinator communities depends on the season of prescribed burns. We show that burning, particularly outside of the winter-dry season, can enhance floral resources for pollinators. Also, the management of a dominant, early-successional plant showed positive effects on flower and pollinator densities, but only in the year following the application of prescribed burns. We thus recommend burning to maximize the abundance and diversity of floral resources for pollinators, and with some inclusion of growing-season (*e.g.*, spring or summer) burns in the fire rotation. Since alternating the season of prescribed burn may benefit other organisms and ecosystem services, we recommend varying burn season, such as alternating growing with dormant season burns. We also recommend a long-term study of plant phenology and nectar production in response to seasonal burn treatments while controlling for dominant early-successional plants or shrubs in fire-adapted ecosystems.

# ACKNOWLEDGEMENTS

The authors thank Scott Sager, Gary Johns, and Gage LaPierre for logistical support and application of prescribed burns. We are grateful to Gage LaPierre, Jesse Frazier, and the Crandall Fire Ecology Lab members for removing yankeeweed and maintaining plots. We acknowledge that this study was conducted in the territory of many natives, including Timucua and Seminole peoples.

## Funding

This project was funded by the University of Florida Foundation, Inc. and McIntire-Stennis Project #FLA-FOR-005759 (to Raelene M. Crandall). The funders had no role in study design, data collection and analysis, decision to publish, or preparation of the manuscript.

## Grant Disclosures

The following grant information was disclosed by the authors:
University of Florida Foundation, Inc.
McIntire-Stennis Project: #FLA-FOR-005759.

## Competing Interests

The authors declare that they have no competing interests.

## Author Contributions

- Opeyemi A. Adedoja analyzed the data, prepared figures and/or tables, authored or reviewed drafts of the article, and approved the final draft.
- Raelene M. Crandall conceived and designed the experiments, performed the experiments, authored or reviewed drafts of the article, and approved the final draft.
- Rachel E. Mallinger conceived and designed the experiments, performed the experiments, analyzed the data, authored or reviewed drafts of the article, and approved the final draft.

## Data Availability

The raw data are available in the Supplemental Files.

## Supplemental Information

Supplemental information for this article can be found online at http://dx.doi.org/10.7717/peerj.14377#supplemental-information.

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
