# Peer review of "Season of prescribed burns and management of an early successional species affect flower density and pollinator activity in a pine savanna ecosystem"

_PeerJ, doi:10.7717/peerj.14377_

## Round 0.1 · original submission · Minor Revisions

Three positive reviews have been received for your submission. All three reviewers have provided valuable suggestions to improve the manuscript by enhancing the research justification in the introduction and including the take-home message for the natural resource managers in conclusion.

Two reviewers commented on the statistical methods, mostly requiring some explanations. For example, what are the differences between Type II and Type III sums of squares and, therefore, the justification for using one instead of the other for determining the significance level for a given comparison?

Please read reviewers' comments carefully and reply how you have addressed reviewers' comments in the revised manuscript (provide a version with tracked changes).

Reviewer 1 ·

Basic reporting

This study explores the effects of fire seasonality on flower density and pollinator diversity in a young longleaf pine forest in Florida. To my knowledge, this is one of the first pollinator studies to address these questions anywhere in the world. I therefore read it hungrily as I am sure many others will once it is published. On top of the fire seasonality question, the researchers also looked at the impacts of removing a weedy native plant from these early successional stands.

Overall, I think it is a very well conducted study. I feel the data on plants are especially strong. Pollinators were mostly categorized into coarse groupings but a major strength is that all flower-visiting insects were included in the analysis- a rarity for such studies.

I don’t see any major flaws in the sampling design or the analysis but I do have a few questions or suggestions that will hopefully help improve the paper:

The biggest problem that stood out to me concerns how the main findings were worded in the abstract, results and discussion sections. For example, one of the most important findings to convey is that there were no differences in pollinator visitation rates between burn treatments in 2019 or 2020. But in the text (e.g., lines 34-37 in the abstract but also in subsequent sections) the authors often give the impression that one season stood out among the others. For example “…and pollinator activity were highest in the summer-wet season fire treatment [in 2019], while…[in 2020] pollinators were most frequent in winter-dry and spring season fire treatments”. I think such results need to be more carefully worded. For example, it would be better to say something like this about pollinator visitation rate in 2019: “Pollinator visitation rate was significantly higher in both the spring and summer-wet fire season treatments compared to the unburned plots but there were no significant differences among the three fire seasons”. To put it more succinctly, I don’t think differences between means should be reported as differences without statistical significance. There are some intriguing patterns in the data that may have been found to be significant with greater replication. But I don’t think they should be given too much weight here considering there were only 3 replicates.

I think the paper would benefit a lot from some description of the nature of the burns during the different seasons. In my experience it seems that growing-season burns are a lot more patchy than dormant season burns. Was this observed in the current study? I am assuming that efforts were made to ensure that the subplots were uniformly burned for each sampling period but it would be good to state this explicitly as well as whether or not fire crews found it difficult for fire to carry during the growing season.

I may have missed it but I didn’t see a figure showing the plant-pollinator network mentioned in lines 247-248.

Please check the text and legends for consistency. For example it seems that pollinator activity and pollinator frequency are used interchangeably but this causes some confusion to readers less familiar with the study.

I found it interesting that time since fire had a positive effect on both flower density and flower richness. This seems somewhat inconsistent with the finding that flower and pollinator numbers trended lower in the unburned plots. However, it is consistent with past studies suggesting that frequent burns may be detrimental to pollinators. There are even some findings from longleaf pine suggesting this. I think the fact that all flower-visiting insects were sampled in this study suggests that the larger pollinator community may be more sensitive to fire than widely believed. It would help to explore this more fully in the discussion.

I think the experimental removal of yankeeweed adds a lot of interest to the study but I feel the rationale for removing it is not very strong. It seems that this is an early successional native species so I wonder why it is viewed as problematic. Won’t it eventually be displaced by later-successional plant species? Also, it would be nice to describe the value of this species to pollinators either based on previous work or on the results from this study.

What should be the main take-away for managers from this study? It seems particularly noteworthy that spring burns resulted in higher pollinator visitation rates than unburned plots in both years (the only treatment for which this was the case). Also, the positive relationship between time since fire and pollinator numbers seems to suggest that burning some areas less frequently may be beneficial. I know that wasn’t one of the questions the researchers originally set out to test but this seems like an important result to address more fully.

Experimental design

good (see full comments under basic reporting)

Validity of the findings

valid but text need clarification in places (see full response under basic reporting)

Additional comments

none

Reviewer 2 ·

Excellent Review

This review has been rated excellent by staff (in the top 15% of reviews)
EDITOR COMMENT
This is one of the most thorough and constructive reviews I have received in quite a while. Thank the reviewer for your time and effort! The review report is thorough and detailed and written in a very objective and constructive tone. Reviewer examination of the experimental design and statistics are very valuable not only for the authors to improve this paper but also for the associate editor to make an overall assessment and decision of the validity of the findings. When the reviewer identified the weakness or provided suggestions for change, the reviewer also provided references or examples, which gives the author a road map and, in most cases, will motivate the author to revisit their work, leading to an overall improvement of the paper. I particularly like the insight from the reviewer regarding beefing up the research justification(s). Those are great suggestions; I believe that incorporating them will significantly improve this paper. Most importantly, by doing so, the author would strengthen writing skills for future publication, especially for writing a grant proposal.

Basic reporting

The paper is overall well-structured and straightforward to read. Background information presented is almost always adequately cited (please see annotated PDF to see a few places where additional citations are recommended). The research presented is self-contained and most of the analysis and results are relevant and sufficient to test predictions. Figures and tables are of sufficient quality. In some places additional detail is needed, as I describe later in my review. Minor revisions to improve basic reporting are as follows:

1. The research manuscript template for PeerJ includes a conclusions section, which is absent from the current manuscript. In addition to conforming to the recommended template, separating the final paragraphs of the discussion under a conclusions header would be a good way to concisely summarize the most important takeaways and recommendations for future work and management applications.
2. The phrase “fire season” is frequently used but could be confusing to some readers as some will associate this phrase with the season in which wildfires are most prevalent rather than the season selected for prescribed burning. Using a different description such as “timing of prescribed burn” or specifically defining the use of “fire season” in the context of this paper could remedy this.
3. In line 352 the phrase “flowering rate” is used but whether this refers to a rate (change in floral density over time), a phenological shift in peak bloom, or simply an increase in floral density is not clear. Using consistent terminology with the rest of the paper would clarify this.
4. It is unclear to me whether/how “pollinator frequency”, a response variable in Table 1, differs from “pollinator density”, “pollinator activity”, or “visitation rate” throughout the paper. If all these refer to the same thing, using consistent terminology would make this clear. If they mean different things then the differences should be explained explicitly.

Experimental design

The topic fits within the aims and scope of PeerJ, and the research questions addressed in the manuscript are relevant and meaningful.

The research question and predictions are clearly stated, but improvements are recommended to the sections dealing with identification of the knowledge gap and framing of predictions:
1. The specific knowledge gap addressed and the unique contributions of the paper could be articulated more clearly in the second paragraph (lines 56-68). The authors begin to identify the key knowledge gaps addressed by their paper by stating that the impact of fire season on pollinator communities has been understudied. However, they then go on to describe several studies which compare the effects of fire season on plant and insect communities, and it is not clear what additional insight is expected from the present study. Comparing the context of the previous studies to this one may help to clarify this point. For example, it appears that most of the studies cited in this section were conducted in prairie ecosystems, and relatively few studies of prescribed burn effects on pollinators have been conducted in forested ecosystems where fire is at least equally important for maintaining pollinator habitat. Highlighting the potential importance of these ecosystems for pollinators could strengthen the justification for the study. Alternatively, if the primary novelty of this study is the examination of the interaction between fire season and weed removal it may be advisable to place more emphasis on the paucity of knowledge in that specific area rather than with regard to the effects of burn season alone.
2. Related to the point made above, the sentence beginning “Whole community responses..” in line 101 seems out of place. This point should be brought up earlier, in the preceding paragraphs that review existing literature and identify the knowledge gap of interest.
3. The logic for the prediction that prescribed burns implemented during the lightning season (lines 105-106) will have the strongest positive effect on plants and pollinators needs further explanation-what mechanism specifically is supposed to produce this result?

The replicated, randomized experimental design used allows for strong inference and is a major strength in this research, as is the frequency of field data collection (every 2 weeks) which allows assessment of continuous change over time.

The authors should provide specific justification for why timed, non-lethal observations were used for pollinator surveys instead of netting with lethal sampling, which would allow higher taxonomic resolution. They should also describe clearly what type of diversity pollinator richness represents in this context given the relatively coarse level of identification used and defend their reasoning for why it is adequate to answer the research question. Are these measures expected to provide a reasonable proxy for species richness? Functional richness? Based on the data attached for review, it seems that the maximum value for pollinator richness is 9-is that sufficient information to meaningfully assess diversity? If so, the authors should provide evidence to support this assumption, and if not they should explain why this is not a major concern or exclude analyses of pollinator richness from the manuscript.

Additionally, I found the following points confusing:
o The purpose of lethally capturing some specimens and identifying them using keys is not clear to me, and it is also not clear to what taxonomic level these specimens were identified. Were these specimens used later in the analysis, and if so, where?
o Why was only the 3-way interaction tested in the model where yankeeweed was subtracted from overall floral density? Why are julian date and time since burn not included?
o There is no explanation for the inclusion of bipartite plots in this analysis-they don’t seem to relate to the research question so I recommend either removing them or explaining how they help to answer the research question.

Validity of the findings

The statistical methods on the whole appear to be thoughtfully chosen and adequate for the purposes of this paper. The most significant issue I identified in the results is that effect sizes are not clearly presented alongside statistical test results, and it is difficult to evaluate the practical importance of the findings without estimates of the magnitude of the differences among groups being compared. Inclusion of effect sizes is strongly encouraged to emphasize biological findings over statistical information (see Nakagawa and Cuthill 2007, Biological Reviews 82, pp 591-605). It is currently possible to make some estimates of effect size from figures, but I strongly recommend that effect size estimates and confidence intervals be included in the text or tables in the results section. The biological relevance of these estimates should also be interpreted in the discussion section to make it clear to the reader whether the differences among treatments is truly appreciable in the context of restoration and biodiversity conservation.

In addition, I would suggest the following minor revisions to the methods, results, and discussion/conclusion:
• Normality assumptions for linear models apply to residuals, not raw data- lines 227-228 should be revised to reflect this.
• Authors do not mention checking distribution of residuals for GLMMs, which is recommended (the DHARMa package may be useful for these purposes).
• I recommend including a brief explanation or citation supporting why Type II and Type III sums of squares are used in different circumstances (lines 230-233).
• The discussion section provides a thoughtful overview of the study findings, considerations of their implications for management, and potential areas for future study. However, in the concluding paragraph (lines 406-417) the language is quite general and primarily emphasizes points that were already supported by prior research (i.e. that fire is an important management tool but there is insufficient information on factors which mitigate its effects) and the value of the present study (i.e. that it shows how the benefit of fire can be maximized) is stated very broadly. I would like to see more emphasis on the concrete findings of this study in this section (e.g. which burn season delivered the largest benefit, that removal of yankeeweed was beneficial but only for a limited period) with recommendations for applications of these findings and suggestions for future research for managers as well as other researchers. This is the part of the paper that I think would be well suited for the conclusions section as mentioned in my “Basic Reporting” comments.

Additional comments

Please see my annotated copy of the manuscript to show minor suggestions to improve clarity and flow of writing.

Annotated reviews are not available for download in order to protect the identity of reviewers who chose to remain anonymous.

Reviewer 3 ·

Basic reporting

The manuscript is relatively clear, well-structured, and covers the literature well in the introduction and discussion. I have no major concerns with “Basic Reporting”. There are a few typos in the figures and some minor critiques to those, which I address in my “General Comments”.

Experimental design

The experimental design and execution is sufficiently explained and well-aligned with the goals of the project. I suggest, in General Comments, some ways in which it could be better communicated (as it is a rather involved design with nested treatments and multiple years) but overall it is appropriate.

Validity of the findings

Conclusions are well-supported by the data. The data are provided in a clean and useful way – however, some sort of metadata file or columns definitions provided elsewhere would be helpful.

Additional comments

This manuscript presents the findings of a study concerned with the impact of prescribed burning and yankeeweed removal on plant-pollinator abundance, richness, and interactions. Overall, I found the work to be fairly well-reported with clear connections between the results and discussion. My main concerns are fairly minor and are centered around explanation of the study design and the description of the value of yankeeweed to pollinators itself.

Major Comments:
• Value of yankeeweed to pollinators: It might be worth a note in the introductory paragraph about yankeeweed whether the plant itself is “useful” to pollinators (lines 84-92). In the results, it is discussed as being visited in “significant” numers (line 263-266). Knowing if the plant itself has value to pollinators helps contextualize what exactly its removal means.

• Study design reporting: The study design is fairly involved with variety of treatments, nesting, and variation of what is evaluated across years. Additionally, because things like burning occur during the experiment, it’s really important that the timing of disturbance is understood by readers. It was also difficult to determine how far control plots are from burned plots (i.e. is there spillover we might expect among plots?). My understanding is they are directly adjacent, but it would be worth making it clear. Perhaps some sort of “timeline” figure would be good.


Minor Comments:
• In abstract wondering if “particularly in late summer and fall” refers to the timing of plants/pollinators or the timing of the burn
• Line 105 maybe add “…flower density and diversity compared to controls….” To clarify greater than what.
• Figure 5 – “Unburned” is misspelled
• Figure 3 – the plotted values represent results from GLME but the points simply represent the mean for each treatment, yes?
• Really minor but Julian Date is not what is used here. It is simply Day of Year (see: https://www.azandisresearch.com/2020/01/27/julian-date-vs-day-of-the-year/)
• Great first paragraph of discussion
• Figure 1 is kinda a lot. I think a more generic sampling scheme figure might be more informative. If kept, what do the points represent? It’s pretty busy so knowing which part goes to which axis/information is important
• Figure 2 – I’m a little confused how flower density would increase throughout the year and then just “collapse” on January 1st (I recognize the graph cuts off here, but presumably if it’s accumulating throughout the year it would “ramp down” after Jan 1, not just end immediately). Is this an artifact of some sort of analytical choice? It seems off
• In stats methods referring to “treatment 1” “treatment 2” is confusing – just call it the effect’s name
• I do not understand what “Statistical significance for each fixed effect 231 was determined with Type II sum of squares ANOVA when interactions were not significant 232 and Type III sum of squares when they were significant.” means

---

## Round 0.2 · Minor Revisions

I invited the original reviewers for a second round of review and only one reviewer is able to provide some further comments. Those are mostly minor comments and require some clarification only. After that, this manuscript is ready for publication.

Reviewer 2 ·

Basic reporting

The basic reporting is adequate and the authors' responses to reviewer suggestions has strengthened the manuscript, particularly in the introduction where the background and study justification are more closely tied together.

Experimental design

Overall study design is strong as I commented in my initial review. The additional details added to the explanation of the pollinator activity sampling approach (using observational, non-lethal methods) is helpful. The rewording in the description of model terms in the statistical analysis section is also clearer than in the initial submission.

Validity of the findings

I appreciate that the authors have added effect size statistics here, however the values reported seem to contradict the statements for which they provide evidence. For example, on line 211 the authors state that summer-wet season treatments had significantly higher floral density than unburned controls, but the confidence interval for Cohen's d includes both negative and positive values (lower bound = -0.17, upper bound = 0.37). I would interpret that confidence interval to mean that the range of plausible values for difference in mean floral density between the treatment and control includes 0, (no difference in means). All the other statements of results in this section that include effect size estimates are similarly confusing. It's possible that this is a formatting issue (like maybe in the example on line 211 the lower bound is -0.37 and the upper bound is -0.17) but whatever the case this should be clarified before acceptance.

I additionally recommend stating more readily interpretable, ideally unstandardized effect size estimates expressed on their original scale for both significant and non-significant results (e.g. as discussed in Nakagawa and Cuthill 2007,which the authors cite, pgs 595-596). For example, it would be useful to know what difference in mean flower density, flower richness and pollinator activity is between treatments in original units of measure (such as blooms/m, # species, visits/minute) expressed with 95% CI following the point estimate. This formulation of results would make the practical significance of the treatment effects easier to interpret than Cohen's d, although Cohen's d may still be useful to include as such standardized effect size statistics facilitate incorporation of results into meta-analyses.

Also, although this is really minor, I find the formatting of the confidence intervals a bit odd as the percentile is not reported and evidence/references to tables and figures provided in parentheses is sometimes separated by commas and sometimes by semi-colons.

Additional comments

Line 41: I think "pollinator activity", the phrase used earlier in the abstract, reads better here than "activities" and I don't think that activity needs to be pluralized to be grammatically correct-the same applies in several places throughout the manuscript.

Line 100: nix the word "generally" unless there is some information on pollination mechanism/pollinator rewards

Line 284: what constitutes "significant visits"? Were these species more commonly visited in 2020 than the species listed as most commonly visited in both years? I recommend rewording to be more explicit about why visitation to these species is of interest.

Line 286: Dalea pinneata is called D. pinneata in line 327, use consistent formatting after the first mention.

Lines 294-295: The sentence "Thus, burn treatments were also more different from one another later in the season" does not clearly follow from the preceding sentence (as implied by the adverb "thus") which seems to be a statement about bloom phenology that is generalized across treatments, so I recommend rewording.

Line 295: Add comma after "Furthermore"

Figure captions: I think that "at P<0.05" should be "at alpha = 0.05".

Figures 2 & 3 y axis titles are a little confusing- "per 5-min per plot" seems like a description/unit given for the rate, so maybe that should be in parentheses, or if the units are # pollinator visits/5 min/plot then I think you could list that information without reiterating that it's a rate.

---

## Round 0.3 · accepted · Accept

I have assessed the revision and confirm that the authors have addressed all comments from the reviewer. This manuscript is ready for publication.